# A Probabilistic Approach to Self-Supervised Learning using Cyclical Stochastic Gradient MCMC

## Abstract

In this paper we present a practical Bayesian self-supervised learning method with Cyclical Stochastic Gradient Hamiltonian Monte Carlo (cSGHMC). Within this framework, we place a prior over the parameters of a self-supervised learning model and use cSGHMC to approximate the high dimensional and multimodal posterior distribution over the embeddings. By exploring an expressive posterior over the embeddings, Bayesian self-supervised learning produces interpretable and diverse representations. Marginalizing over these representations yields an improvement in performance, calibration and out-of-distribution detection in downstream task. We provide experimental results on multiple classification tasks on five challenging datasets. Moreover, we demonstrate the effectiveness of the proposed method in out-of-distribution detection using the SVHN dataset.

## 1 Introduction

Self-supervised learning is a learning strategy where the data themselves provide the labels (Jing & Tian (2020)). The aim of self-supervised learning is to learn useful representations of the input data without relying on human annotations (Zbontar et al. (2021)). Since they do not rely on annotated data, they have been used as an essential step in many areas such as natural language processing, computer vision and biomedicine(Jospin et al. (2020)).

Contrastive methods (Chen et al. (2020)) are one of the promising self-supervised learning approaches which learn representations by maximizing the similarity between embeddings obtained from different distorted versions of an image (Zbontar et al. (2021)). Several tricks are proposed to overcome the issue of feature collapse. These include using negative samples in simCLR (Zbontar et al. (2021)) and stop gradient in BYOL (Grill et al. (2020)).

Self-supervised models are often trained using stochastic optimization methods which approximate the *distribution* over the parameters with a *point mass* ignoring the uncertainty in the parameter space. Indeed if the regularizer imposed on the model parameters is viewed as the the log of a prior on the distribution of the parameters, optimizing the cost function may be viewed as a maximum *a-posteriori* (MAP) estimate of model parameters (Li et al. (2016)). Bayesian methods provide principled alternatives that model the whole posterior over the parameters and account for model uncertainty in the parameter space (Zhang et al. (2020)).

Exact Bayesian learning of Deep Neural Networks is generally intractable, hence Bayesian deep learning models use approximation methods like variational inference (VI, Blundell et al. (2015)) or MCMC methods (Neal (2012)) to capture the posterior over the parameters and estimate model uncertainty. While VI methods usually approximate a single mode, MCMC methods are used for sampling from different modes (Jospin et al. (2020)). In a recent line of work, Stochastic Gradient Markov Chain Monte Carlo (SG-MCMC) methods (Welling & Teh (2011), Chen et al. (2014), Ma et al. (2015)) were proposed which couple MCMC with SGD to provide a promising sampling approach to inference in Bayesian deep learning for large datasets (Welling & Teh (2011)). In another work, Zhang et al. (2020) proposed Cyclical Stochastic Gradient MCMC (cSG-MCMC) in order to explore a highly multimodal parameter space given a realistic computational budget (Zhang et al. (2020)).

In this paper we aim to adapt Bayesian supervised learning concepts to self-supervised learning to make a self-supervised learning model fully probabilistic using cSG-MCMC. Our motivation comes from the fact that the posterior distribution over the parameters of a self-supervised learning model may be multimodal and thus insufficiently represented by a single point estimate. By exploring the posterior distribution over the parameters instead of point mass we aim to improve performance in downstream tasks. Moreover, the optimization step in cSG-MCMC involves injection of Gaussian noise to the parameter update of SGD which helps to alleviate the feature collapse issue in the contrastive methods and make the features more informative (Li et al. (2016)).

In this paper, we propose a simple Bayesian formulation for self-supervised learning with a specific family of cSG-MCMC methods called Cyclical Stochastic Gradient Hamiltonian Monte Carlo (cS-GHMC) (Zhang et al. (2020)). Within this framework, we use BYOL as a self-supervised learning model which allows to incorporate Bayesian learning of an approximate posterior over the parameters instead of MAP. Our experimental results indicate that by integrating a Bayesian learning we can achieve better performance in downstream tasks including classification and out-of-distribution detection. The simplicity of the proposed approach is one of its greatest strengths.

## 2 PROBLEM STATEMENT

Given a dataset $\mathcal{D}$, a self-supervised learning model $\mathcal{F}_\theta$ parameterized by $\theta$, aims to produce a representation $Z_\theta$ by solving a predefined proxy task. In this paper we wish to learn a distribution over the embeddings $Z_\theta$ by placing a prior over the parameters $\theta$ and using Bayesian learning instead of MAP estimation. For learning the representations we use BYOL, a recent self-supervised learning method based on contrastive learning. To obtain the distribution over the embeddings, we use cSGHMC. In the following, first we describe the self-supervised learning model to learn representations. Then, we describe cSGHMC and highlight how it allows to obtain a distribution over the embeddings.

### 2.1 SELF SUPERVISED LEARNING

The aim of contrastive learning is to learn representations by contrasting two augmented views of an image. Particularly BYOL learns representations by reducing a contrastive loss between two neural networks referred to as online network $\mathcal{F}_\theta$ (parameterized by $\theta$) and target network $\mathcal{F}_\xi$ (parameterized by $\xi$). Each network consists of three components, an encoder $f(.)$ (e.g., Resnet-18), a projection head $g(.)$ (e.g., an MLP) and a prediction head $q(.)$ (e.g., an MLP). For a given mini-batch $X = \{x_i\}_{i=1}^N$ sampled from a dataset $\mathcal{D}$ it produces two distorted views, $t(X)$ and $t'(X)$, via a distribution of data augmentations $\mathcal{T}$. The two batches of distorted views then are fed to the online network and the target network, producing batches of embeddings, $Z_\theta$ and $Z_\xi$, respectively. These features are then transformed with the projection heads into $Y_\theta$ and $Y_\xi$. The online network then outputs a prediction $Q_\theta$ of $Y_\xi$ using the prediction head $q_\theta(.)$. Finally the following mean squared error between the normalized predictions $\bar{Q}_\theta$ and target projections $\bar{Y}_\xi$ is defined:

$$\mathcal{L}_{\theta,\xi} = \|\bar{Q}_\theta - \bar{Y}_\xi\|^2 = 2 - 2.\frac{\langle \bar{Q}_\theta, \bar{Y}_\xi \rangle}{\|\bar{Q}_\theta\|.\|\bar{Y}_\xi\|}. \tag{1}$$

$\tilde{\mathcal{L}}_{\theta,\xi}$ is computed by separately feeding $t'(X)$ to the online network $\mathcal{F}_\theta$ and $t(X)$ to the target network $\mathcal{F}_\xi$. Then, at each training step, a stochastic optimization step is performed to minimize

$$\mathcal{L}_{\theta,\xi}^{\text{BYOL}} = \mathcal{L}_{\theta,\xi} + \tilde{\mathcal{L}}_{\theta,\xi} \tag{2}$$

The gradient is taken only with respect to $\theta$. So, the parameter update is as follows:

$$\theta \leftarrow \text{optimizer}(\theta, \nabla_\theta \mathcal{L}_{\theta,\xi}^{\text{BYOL}}). \tag{3}$$
$$\xi \leftarrow \tau\xi + (1-\tau)\theta,$$

where the weights $\xi$ are an exponential moving average of the online network's parameters $\theta$ with a target decay rate $\tau \in [0,1]$. At the end of training, the encoder $f_\theta(.)$ is used for the downstream task. During training only the parameters $\theta$ of the online network $\mathcal{F}_\theta$ are updated.

## 2.2 Posterior Sampling using cSGHMC

In the Bayesian paradigm, for a given dataset $\mathcal{D} = \{x_i\}_{i=1}^n$ and a $\theta$-parameterized model, the following *a-posterior distribution* over $\theta$ is computed using Bayes' rule as: $p(\theta|\mathcal{D}) \propto p(\mathcal{D}|\theta)p(\theta)$, where $p(\theta)$ is a *prior* assigned to the parameters $\theta$ and $p(\mathcal{D}|\theta)$ is the likelihood.

In MAP optimization, the prior has the role of a regularizer and the likelihood has the role of a cost function. An optimizer is optimized to find the MAP solution which is amenable to the parameter update:

$$\Delta\theta = -\frac{\ell}{2}\left(\frac{n}{N}\sum_{i=1}^{N}\nabla_\theta\log p(x_i|\theta) + \nabla_\theta\log p(\theta)\right), \tag{4}$$

for a given randomly sampled mini-batch $X = \{x_i\}_{i=1}^N \subset \mathcal{D}$ and learning rate $\ell$.

In contrast to MAP optimization, in the Bayesian paradigm the model explores the distribution over the model parameters. Welling & Teh (2011) showed that this distribution can be approximated using Stochastic Gradient Langevin Dynamics (SGLD) by injecting Gaussian noise to the parameter updates of SGD so that they do not collapse to just the MAP solution. This leads to the following parameter update:

$$\Delta\theta = -\frac{\ell}{2}\left(\frac{n}{N}\sum_{i=1}^{N}\nabla_\theta\log p(x_i|\theta) + \nabla_\theta\log p(\theta)\right) + \epsilon; \quad \epsilon \propto \mathcal{N}(0, \ell I) \tag{5}$$

SGHMC (Chen et al. (2014)) is an improved counterpart of SGLD which introduces a momentum variable $\boldsymbol{m}$. The posterior sampling is done using the following update rule:

$$\boldsymbol{m} = \beta\boldsymbol{m} - \frac{\ell}{2}\left(\frac{n}{N}\sum_{i=1}^{N}\nabla_\theta\log p(x_i|\theta) + \nabla_\theta\log p(\theta)\right) + \epsilon; \quad \epsilon \propto \mathcal{N}(0, (1-\beta)\ell I) \tag{6}$$

$$\theta = \theta + \boldsymbol{m}$$

where $\beta$ is the momentum term. Equations 5 and 6 guarantee convergence to the true posterior as long as the learning rate $\ell$ decreases towards zero. Zhang et al. (2020) showed that replacing the traditional decreasing learning rate schedule in SGHMC with a cyclical variant allows to explore multimodal posterior distributions and developed cSGHMC. In this paper we apply cSGHMC to take samples from the posterior distribution.

## 3 Posterior over representations

To infer a posterior over the embeddings, we place a prior $\alpha$ over the parameters $\theta$ of the online network $\mathcal{F}_\theta$. By placing a distribution over $\theta$, we induce a distribution over an infinite space of online networks $\mathcal{F}_\theta$. This results in a distribution over embeddings $Z_\theta$. Sampling from this distribution corresponds to sampling from the following conditional posterior:

$$Z_\theta \propto p(\theta|X) \propto p(X|\theta)p(\theta|\alpha), \tag{7}$$

where $X$ is a mini-batch. Equation 7 can be interpreted intuitively as follows. We sample weights $\theta$ from the prior $p(\theta|\alpha)$. Then we condition on this sample of the weights form a particular online network $\mathcal{F}_\theta$ which is used to produce embedding $Z_\theta$ by minimizing the loss function $\mathcal{L}_{\theta,\xi}^{\text{Byol}}$.

To take samples from the posterior in equation 7 we use cSGHMC and the update rule in equation 6. Algorithm 1 describes our proposed method to sample from the posterior. Algorithm 1 produces samples from the posterior over the parameters $\theta$ of the online network $\mathcal{F}_\theta$ and consequently a distribution over embeddings $Z_\theta$, since we compute the gradients of the loss with respect to the parameters that we are sampling.

Our proposed probabilistic approach is a natural Bayesian generalization of MAP optimization. Indeed, if one performs MAP optimization using SGD in Algorithm 1 instead of posterior sampling, one approximates the whole posterior over $\theta$ with a point mass. Sampling from the whole posterior over embeddings versus approximating this with a point mass allows estimating uncertainties in the embedding space.

---

**Algorithm 1** Probabilistic Self-Supervised Learning

---

**Require:** $\ell$ initial learning rate, $\beta$ momentum term, $K$ number of training iterations
**Ensure:** sequence $\theta^1, \theta^2, ...$
   **for** $k = 1 : K$ **do**
       • sample mini-batch $X = \{x_i\}_{i=1}^N$ and augmentations $t \sim \mathcal{T}, t' \sim \mathcal{T}$
       • compute $\mathcal{L}_{\theta,\xi}^{\text{BYOL}}$ based on equation 2
       $\ell \leftarrow C(k)\ell$                      $\triangleright$ update learning rate using cyclic modulation
       $\boldsymbol{m} \leftarrow \beta\boldsymbol{m} - \dfrac{\ell}{2}\nabla_\theta \mathcal{L}_{\theta,\xi}^{\text{BYOL}} + \epsilon; \;\; \epsilon \propto \mathcal{N}(0, (1-\beta)\ell I)$
       $\theta \leftarrow \theta + \boldsymbol{m}$                   $\triangleright$ update parameters $\theta$ using update rule 6
       $\xi \leftarrow \tau\xi + (1-\tau)\xi$
       **if** end of cycle **then**
          yield $\theta$
       **end if**
   **end for**

---

**Marginalizing over representations:** Once pretraining is done, we can marginalize the posterior over $\theta$ for downstream tasks. To compute the predictive distribution for a new instance $x^*$ we use a model average over all collected samples with respect to the posterior over $\theta$.

$$p(y^*|x^*, \mathcal{D}) = \int p(y^*|x^*, \theta)p(\theta|\mathcal{D})d\theta \approx \frac{1}{T}\sum_{t=1}^{T} p(y^*|x^*, \theta^{(k)}), \theta^{(k)} \propto p(\theta|\mathcal{D}). \quad (8)$$

We will see that this model average is effective for improving performance, calibration as well as out-of-distribution detection in downstream task.

## 4 EXPERIMENTS

In this section, we present our experimental results. We evaluate proposed method on several tasks including semi-supervised learning and out-of-distribution detection. First, we describe our experimental setup. Then, we evaluate our model using semi-supervised setting, and lastly, we evaluate our model using out-of-distribution examples. We implemented the code using PyTorch (Paszke et al. (2017)). Our code is available upon request.

### 4.1 EXPERIMENTAL SETUP

**Datasets** We conduct our experiments on five challenging image datasets. A brief description of these datasets is summarized in Table 1. Both the training and test set are used for CIFAR-10, CIFAR-100 (Krizhevsky & Hinton (2009)), and STL-10 (Coates et al. (2011)), while only the training set is used for ImageNet-10 (Chang et al. (2017)), and Tiny-ImageNet (Le & Yang (2015)). For Imagenet-10 and Tiny-ImageNet the Validation set of Imagenet-10 is used for evaluation, since the Test set of these datasets do not have ground-truth. For STL-10, its 100,000 unlabeled samples are used for pretraining.

**Implementation Details** We adopt ResNet-18 (He et al. (2016)) as the backbone for the self-supervised learning model. Following the original setting of BYOL, we use 2-layer MLPs as the

Table 1: A summary of datasets used for evaluations.

| Dataset | Pretrain | Fine-tune | Test | Samples | Classes | Image Size |
|---------|----------|-----------|------|---------|---------|------------|
| CIFAR-10 | Train | Train | Test | 60000 | 10 | 32 |
| CIFAR-100 | Train | Train | Test | 60000 | 100 | 32 |
| STL-10 | Unlabeled | Train | Test | 113000 | 10 | 96 |
| ImageNet-10 | Train | Train | Validation | 13000 | 10 | 224 |
| Tiny-ImageNet | Train | Train | Validation | 103925 | 10 | 64 |

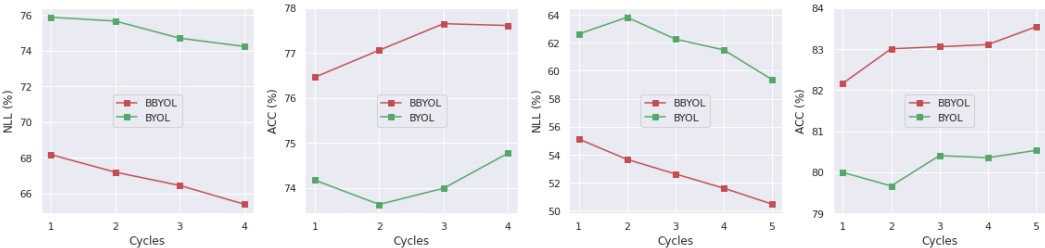

Figure 1: NLL and ACC in downstream task for embeddings of last 4 cycles of both models. **left**: Results on CIFAR-10, **right**: Results on ImageNet-10. Embeddings from BBYOL show consistanly lower NLL and higher ACC.

projection and prediction heads. We apply the standard ResNet without modification on the input images of given sizes in Table 1 for all datasets which produces a feature vector of size $512$ for each sample. We use terms "representations" or "embeddings" interchangeably for this feature vector. We use the same set of data augmentations in Grill et al. (2020) on all datasets for pretraining, consists of random cropping and resizing with a random horizontal flip, followed by a color distortion and a grayscale conversion.

For cSGHMC, a normal prior $\mathcal{N}(0, I)$ is set on the parameters of the online network and a cosine decay learning rate (Zhang et al. (2020)) with cycle length of $50$ is used as a scheduler. The batch size is set to $256$ and we use momentum term of $0.9$. For all datasets we train the model from scratch for 1000 epochs with the initial learning rate $0.1$ for Tiny-ImageNet and $0.2$ for the rest of datasets. We inject Guassian noise at epoch $45$ and collect 1 sample per cycle for the last 4 cycles resulting in 4 samples. We scale Guassian noise and prior with the dataset size (Florian et al. (2020)).

**Evaluation Metrics**  Tow widely-used metrics including Accuracy (ACC), and Negative Log Likelihood (NLL) are utilized to evaluate our method. Higher value of ACC indicate better performance of the model and lower value of NLL indicate better calibration.

**Baselines** We compare proposed Bayesian self-supervised model referred as BBYOL with its deterministic counterpart in terms of described evaluation metrics. For training BYOL we use SGD optimizer with a fixed learning rate schedule and momentum term $0.9$. No weight decay is used. Other parameters are as the same as BBYOL. The initial learning rate for each dataset is described in Appendix A. We train BYOL on all datasets for 1000 epochs and take 4 samples on last 200 epochs with the same interval 50 epochs. For both BYOL and BBYOL if we marginalize over representations we use terms BYOL-ENS and BBYOL-ENS respectively.

The experiments are carried out on Nvidia A40 48 GB and it takes about 7 gpu-hours to train the model on CIFAR-10 and CIFAR-100, 21 gpu-hours on STL-10, 9 gpu-hours on ImageNet-10, 24 gpu-hours on Tiny-ImageNet. We repeat experiments for 3 random seeds and report average NLL and ACC over 3 runs with the standard error from the mean predictor.

### 4.2    IMAGE CLASSIFICATION

We evaluate the performance obtained when fine-tuning both BYOL and BBYOL's representations on a classification task. In this task a pretrained model is fine-tuned on subsets of original training dataset with labels. We fine-tune on $100\%$ of labeled training data described in Table 1. We follow protocol in Grill et al. (2020) detailed in Apendix B. We report both ACC and NLL on five challenging datasets in Table 2 and Table 3. According to the results shown in Table 2 and 3, BBYOL significantly outperforms BYOL and BYOL-ENS on all five datasets. In particular, BBYOL improves NNL on all datasets by large margin compared to its deterministic counterpart. Moreover, marginalizing over the representations in BBYOL-ENS also improves performance in terms of both ACC and NLL. For STL-10 and Tiny-ImageNet we marginalize over two embeddings of last two cycles.

Table 2: ACC $\uparrow$ and NLL $\downarrow$ on five object image benchmarks. The best results are shown in boldface.

| Dataset | CIFAR-10 | | CIFAR-100 | | STL10* | |
|---|---|---|---|---|---|---|
| Metrics | ACC | NLL | ACC | NLL | ACC | NLL |
| BYOL | $74.0 \pm 0.0$ | $72.3 \pm 0.4$ | $44.6 \pm 0.09$ | $210.0 \pm 0.0$ | $65.3 \pm 1.5$ | $96.2 \pm 3.6$ |
| BYOL-ENS | $75.0 \pm 0.0$ | $69.9 \pm 0.8$ | $46.5 \pm 0.2$ | $199.3 \pm 0.4$ | $65.6 \pm 1.7$ | $94.6 \pm 4.4$ |
| **BBYOL** | $\mathbf{76.3} \pm 0.4$ | $\mathbf{66.3} \pm 0.9$ | $\mathbf{45.6} \pm 0.4$ | $\mathbf{202.0} \pm 2.1$ | $\mathbf{77.9} \pm 0.2$ | $\mathbf{60.4} \pm 0.5$ |
| **BBYOL-ENS** | $\mathbf{77.3} \pm 0.4$ | $\mathbf{65.0} \pm 0.8$ | $\mathbf{47.5} \pm 0.4$ | $\mathbf{193.3} \pm 2.3$ | $\mathbf{78.1} \pm 0.2$ | $\mathbf{60.4} \pm 0.6$ |

Table 3: ACC $\uparrow$ and NLL $\downarrow$ on five object image benchmarks. The best results are shown in boldface.

| Dataset | ImageNet-10 | | Tiny-ImageNet* | |
|---|---|---|---|---|
| Metrics | ACC | NLL | ACC | NLL |
| BYOL | $80.1 \pm 0.1$ | $61.0 \pm 1.5$ | $69.8 \pm 1.3$ | $94.1 \pm 4.1$ |
| BYOL-ENS | $80.6 \pm 0.3$ | $58.8 \pm 0.7$ | $71.1 \pm 0.3$ | $89.0 \pm 1.5$ |
| **BBYOL** | $\mathbf{83.0} \pm 0.3$ | $\mathbf{51.3} \pm 1.0$ | $\mathbf{76.3} \pm 0.2$ | $\mathbf{74.3} \pm 0.4$ |
| **BBYOL-ENS** | $\mathbf{83.0} \pm 0.0$ | $\mathbf{50.6} \pm 1.1$ | $\mathbf{76.5} \pm 0.1$ | $\mathbf{72.5} \pm 0.2$ |

We observe that marginalizing over the representations in BYOL-ENS also improves performance. It is due to the nature of contrastive loss which induces diversity in the parameter space. Whenever the loss is not too high, marginalizing over these representations improve the performance.

In Figure 1, we plot ACC and NLL of embeddings obtained from last 4 cycles for CIFAR-10 and ImageNet-10. We observe that BBYOL consistently yields lower error and higher accuracy, demonstrating that the embeddings obtained from a BBYOL are more informative.

**Ensemble Size** In some applications, it may be beneficial to vary the size of the ensemble dynamically at test time depending on available resources. Figure 2 displays the performance of BBYOL on CIFAR-10 and CIFAR-100 datasets as the effective ensemble size, is varied. Although ensembling more models generally gives better performance, we observe significant drops in error when the second and third models are added to the ensemble. In most cases, an ensemble of two models outperforms the baseline model.

### 4.3 OUT-OF-DISTRIBUTION DETECTION

To further explore the diversity of embeddings, we consider the out-of-distribution (OOD) detection task. A pretrained model on CIFAR-10 was fine-tunned on its training data with labels and tested on SVHN dataset (Netzer et al. (2011)); and a pretrained model on CIFAR-100 was fine-tunned on its labeled training data and was tested on SVHN. We estimate the entropy of the predictive distribution on SVHN. Figure 3 (left) presents the empirical CDF of the predictive entropy for CIFAR-10 (in-

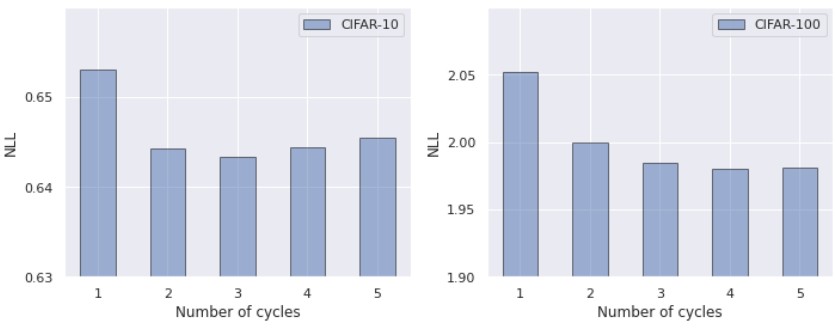

Figure 2: NLL as a function of number of ensembles on CIFAR-10 and CIFAR-100.

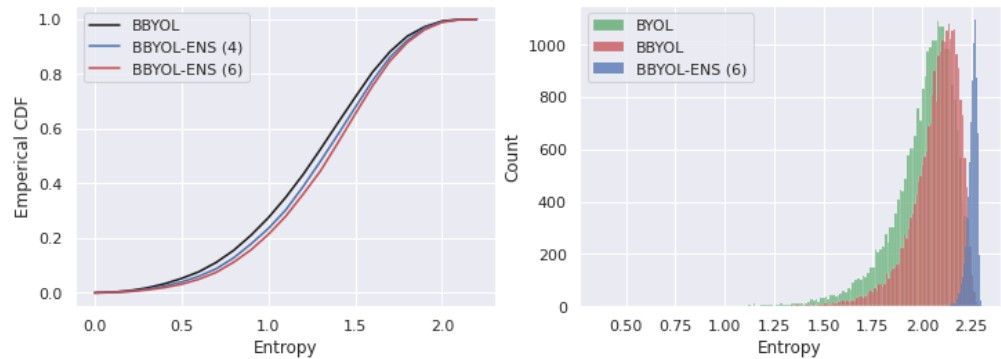

Figure 3: **left**: Empirical CDF for the predictive entropy of SVHN pretrained on CIFAR-10. **right**: Hitogram for the predictive entropy of SVHN pretrained on CIFAR-100. BYOL-ENS (4/6) is marginalizing over 4/6 embeddings.

distribution) and SVHN (out-of-distribution). Figure 3 (right) indicates the histogram of predictive entropy for CIFAR-100 (in-distribution) and SVHN (out-of-distribution).

For the unseen data, we expect the model indicates max entropy and the probablity of low entropy should be lower (Zhang et al. (2020)). It also implies that the mode of histogram focuses at higher value. We observe that all models assign high entropy to unseen data, since self-supervised learning improves model uncertainty (Hendrycks et al. (2019)). In particular the predictive uncertainty improves on unseen classes, as the ensemble size increases. It indicates that the embeddings produced by sampling from the posterior come from different modes and provide different characterization of training data. Indeed different embeddings can provide different predictions on OOD, leading to better uncertainty estimation.

Table 4 summarizes the results for BYOL, BBYOL and BBYOL-ENS with different ensemble size. BBYOL-ENS (6) indicates marginalizing over 6 embeddings collected from last 6 cycles. In BBYOL-ENS (12*) we took 3 samples per cycle in last 4 cycles. Consistent with our previous results BBYOL improves BYOL in terms of calibration (lower NLL) and uncertainty estimation (higher Entropy). The improvement in calibration and uncertainty estimation consistently increases by increasing the number of ensemble size.

Table 4: Results for OOD detection for various settings. Pretrained models are fine-tunned over training labeled data of CIFAR-10 and CIFAR-100 and tested on SVHN. All values are in percentages. ↑ indicates larger value is better, and ↓ indicates lower value is better.

| In-Distribution | Out-of-Distribution | Method | NLL ↓ | AUROC ↑ | Mean-Ent ↑ |
|---|---|---|---|---|---|
| | | BYOL | 5.22 | 47.13 | 0.99 |
| | | BBYOL | 5.12 | 47.24 | 0.98 |
| Cifar-10 | SVHN | BBYOL-ENS (4) | 5.12 | 47.04 | 1.05 |
| | | BBYOL-ENS (6) | **4.99** | **47.82** | **1.10** |
| | | BBYOL-ENS (12*) | 5.02 | 47.74 | 1.09 |
| | | BYOL | 2.62 | 49.75 | 2.0 |
| | | BBYOL | 2.55 | 50.02 | 2.06 |
| Cifar-100 | SVHN | BBYOL-ENS (4) | 2.36 | 49.83 | 2.23 |
| | | BBYOL-ENS (6) | 2.34 | 50.0 | 2.25 |
| | | BBYOL-ENS (12*) | **2.30** | **50.11** | **2.30** |

## 5 DISCUSSION AND FUTURE WORK

In this paper, we propose to explore the distribution over the representations using cSGHMC instead of estimating the posterior with a MAP solution. Our experimental results indicate that samples taken from the posterior provide meaningfully different representations which leads to improved accuracy, calibration and uncertainty estimation in downstream tasks. Moreover through our experimental results we indicate that using cSGHMC instead of a deterministic counterpart improves the quality of representations in downstream task in terms of both accuracy and calibration. Capturing the posterior over the embeddings can also provide the possibility of uncertainty estimation in the embedding space, opening new directions for practitioners in the field.

For future work we would like to further investigate the following: (i) We would like to analyze the notion of uncertainty in the embedding space and its relation with uncertainty in the predictive space. (ii) We would like to explore the possibility of using a Bayesian self-supervised learning model as a prior for a Bayesian downstream task.

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

## A    APPENDIX

For training BYOL using SGD optimizer we set initial learning rate $\ell = 3e - 4$ for STL-10 and Tiny-ImageNet. We set $\ell = 3e - 3$ for the rest of datasets. For cSGHMC, as mentioned in Zhang et al. (2020), tempering helps. We use temperature $0.1$ for all datasets.

## B    APPENDIX

For fine-tunning, we follow the protocol of Grill et al. (2020). We first initialize the network with the parameters of the pretrained representation, and fine-tune it with a subset of original datasets with labels. We do not use any data augmentation during fine-tunning. We optimize the cross-entropy loss using SGD with Nesterov momentum with batch size of $100$, and a momentum of $0.9$. We sweep over the learning rate $\{2e - 5, 1e - 5, 1e - 4, 2e - 4, 3e - 4, 4e - 4, 5e - 4\}$, weight decay $\{0, 5e - 4\}$ and the number of epochs $\{50, 60\}$ and select the hyperparameters achieving the best performance on our local validation set to report test performance. Table 5 describes parameters for each dataset.

Table 5: Parameters used in fine-tunning.

| Dataset | learning rate | weight decay | number of epochs |
|---------|---------------|--------------|------------------|
| CIFAR-10 | $2e-4$ | 0 | 50 |
| CIFAR-100 | $1e-4$ | 0 | 50 |
| STL-10 | $2e-4$ | 0 | 50 |
| ImageNet-10 | $2e-4$ | 0 | 50 |
| Tiny-ImageNet | $2e-4$ | $5e-4$ | 50 |

