# OpenReview forum: "A Probabilistic Approach to Self-Supervised Learning using Cyclical Stochastic Gradient MCMC "
_ICLR.cc/2023/Conference — Submitted to ICLR 2023_

### Official Review · Reviewer_YvPk · 2022-10-17

**Confidence:** 5
**Correctness:** 2
**Technical Novelty And Significance:** 1
**Empirical Novelty And Significance:** 1
**Recommendation:** 3

**Clarity, Quality, Novelty And Reproducibility:**

The clarity and quality of the current manuscript are poor. In fact, I think the manuscript is not completed yet. The novelty is quite limited.
As the proposed techniques are kind of easy, the reproducibility is believed satisfactory.

**Strength And Weaknesses:**

Strength.
(1) The research direction is interesting.

Weaknesses.
(1) The underlying logic and notations are in a mess.
(2) The novelty is quite limited.
(3) The presented techniques are not convincing overall.

**Summary Of The Paper:**

The authors propose introducing Bayesian inference to self-supervised learning for improved model generalization capability. Specifically, they view the traditional learning of the BYOL as a MAP estimation; accordingly, Bayesian posterior sampling methods like the cSGHMC can be leveraged to replace the vanilla SGD optimizer for posterior inference of the model parameters. Experiments on semi-supervised classification tasks are conducted.

**Summary Of The Review:**

(1) As mentioned in the Abstract, a Bayesian self-supervised learning method is proposed. So why use semi-supervised classification tasks to evaluate its effectiveness? Similarly, how would those experiments justify your goal of ``improving performance in downstream tasks,'' as mentioned in the introduction?

(2) In Eq. (7), what are the definitions of the likelihood and the prior, when we talk about a self-supervised learning method? Besides, since X is a mini-batch, Eq. (7) is actually quite confusing; you are not performing cSGHMC on the mini-batch X, right? Similarly, there are other notations that are confusing/wrong.

---

### Official Review · Reviewer_tf9n · 2022-10-24

**Confidence:** 5
**Correctness:** 3
**Technical Novelty And Significance:** 1
**Empirical Novelty And Significance:** 1
**Recommendation:** 3

**Clarity, Quality, Novelty And Reproducibility:**

- The method is easy to follow as it is a combination of techniques. But overall, it feels like this paper is unfinished and quality is poor.

 - Novelty is very minor.

- Not enough information to reproduce their results but I believe since it is an incremental technique, one should be able to reproduce.

**Strength And Weaknesses:**

Strength:
- Rather an important problem.
- It is simple to follow and straightforward.
- Enough background is provided.
-------------------------------------------------------
Weakness:
- This paper is highly incremental meaning it is a direct combination of two things. Using BYOL for representation learning and then using a variation of HMC to sample from the distribution both of which were introduced in other papers.
- All the techniques for predictive distribution approximation and etc are very straightforward and the novelty is very minor.
- Simulation results are extremely insufficient. It would have been useful if a comparison with other state-of-the-art frameworks was provided.

**Summary Of The Paper:**

propose a Bayesian technique by enforcing a prior over parameters for self-supervised learning. The main idea is based on the BYOL to learn the representation and combine that with Cyclical SGHMC. Paper applies the method to two datasets for semi-supervised classification and one dataset for out-of-distribution detection.

**Summary Of The Review:**

Even though the problem is rather important but this paper is highly incremental as it comes down to two basic ideas that were introduced and used by others. Experimental results are insufficient and it does not sufficiently compare this framework with other well-understood methods.

---

### Official Review · Reviewer_6qBF · 2022-10-26

**Confidence:** 4
**Correctness:** 2
**Technical Novelty And Significance:** 1
**Empirical Novelty And Significance:** 1
**Recommendation:** 3

**Clarity, Quality, Novelty And Reproducibility:**

Overall I think the paper is easy to follow. However, I don't see a significant contribution to be considered novel. The authors did not provide detailed information to reproduce experiments or a code.

**Strength And Weaknesses:**

Strengths
- Bayesian self-supervised learning is relatively underdeveloped; the paper is tackling an interesting problem.

Weaknesses
- Limited novelty. I see no contribution other than a mere application of cSGMCMC to BYOL.
- Bayesian inference requires Monte-Carlo approximation which requires multiple forward passes through encoders. This is a critical downside, considering that we typically employ deep neural networks with millions of parameters for encoders. I'm not sure whether the benefit from the performance improvement outweighs the increased inference cost.
- Experiments were done only for relatively small-scale datasets (CIFAR-10 and CIFAR-100).
- When training Bayesian neural networks, one should be careful about using data augmentations, since the resulting model might not be interpreted as a valid Bayesian model. To bypass this technical difficulty, existing works studying BNNs often discard data augmentations; I think training BYOL is even harder than vanilla BNNs to be cast as a Bayesian inference problem, especially due to the use of slow-update parameter $\xi$ (which also is a function of $\theta)$. For instance, given the data augmentation policy and moving average parameter $\xi$, what would be a valid likelihood? This is not an easy question to answer.

**Summary Of The Paper:**

This paper introduces a Bayesian version of the self-supervised learning algorithm (BYOL), where the cyclical stochastic gradient MCMC (cSGMCMC) is employed for approximate posterior inference. Instead of point estimates, the proposed approach constructs posterior distributions of encoder parameters and uses them for downstream tasks. The proposed approach is demonstrated to outperform non-Bayesian baselines on benchmark datasets.

**Summary Of The Review:**

I think in the current form the submission is not ready for publication, mainly due to the fact that it is a straightforward application of cSGMCMC to BYOL without much care.

---

### Decision · Program_Chairs · 2023-01-20

**Decision:**

Reject

**Justification For Why Not Higher Score:**

Consensus among reviewers that the paper was not ready for publication in its current form

**Justification For Why Not Lower Score:**

N/A

**Metareview: Summary, Strengths And Weaknesses:**

All three reviewers agreed that this paper was not ready for publication; in particular, there were concerns that the method was too incremental (with limited novelty) and that the empirical validation was too small-scale to be sufficient.